# Dietary Intakes and Cardiovascular Health of Healthy Adults in Short-, Medium-, and Long-Term Whole-Food Plant-Based Lifestyle Program

**DOI:** 10.3390/nu12010055

**Published:** 2019-12-24

**Authors:** Boštjan Jakše, Barbara Jakše, Stanislav Pinter, Borut Jug, Uroš Godnov, Jernej Pajek, Nataša Fidler Mis

**Affiliations:** 1Department of Nutrition, Biosciences, Biotechnical Faculty, University of Ljubljana, Jamnikarjeva 101, 1000 Ljubljana, Slovenia; bostjanjakse@hotmail.com; 2Barbara Jakše sole proprietor, 1230 Domžale, Slovenia; barbara.tursic@gmail.com; 3Basics of Movements in Sport, Faculty of Sport, University of Ljubljana, Gortanova 22, 1000 Ljubljana, Slovenia; stane.pinter@gmail.com; 4Centre for Preventive Cardiology, Department of Vascular Diseases, University Medical Centre Ljubljana, Zaloška 2, 1525 Ljubljana, Slovenia; borut.jug@kclj.si; 5Faculty of Medicine, University of Ljubljana, Vrazov trg 2, 1000 Ljubljana, Slovenia; 6Department of Computer Science, Faculty of Mathematics, Natural Sciences and Information Technologies, University of Primorska, Titov trg 4, 6000 Koper, Capodistria, Slovenia; uros.godnov@gmail.com; 7Department of Nephrology, University Medical Center Ljubljana, Zaloška 2, 1525 Ljubljana, Slovenia; jernej.pajek@mf.uni-lj.si; 8Department of Gastroenterology, Hepatology and Nutrition, University Children’s Hospital, University Medical Centre Ljubljana, Bohoričeva 20, 1000 Ljubljana, Slovenia

**Keywords:** nutrition, plant-based diet, vegan diet, lifestyle, cardiovascular health, blood lipids, blood pressure

## Abstract

An effective lifestyle strategy to reduce cardiovascular diseases risk (CVD) factors is needed. We examined the effects of a whole-food plant-based (WFPB) lifestyle program on dietary intake and cardiovascular (CV) risk factors in 151 adults (mean 39.6 (SD 12.5) years). Adherence was categorised into short-, medium- and long-term (years: (0.5–<2), (2–<5) and (5–10)), for both genders separately. Dietary intakes were assessed, fasting blood lipids and blood pressure (BP) were measured, and % participants reaching guideline recommended targets for LDL-cholesterol, triglycerides and BP in the primary CVD prevention was assessed. There were no statistically significant differences in intakes of energy and most nutrients among participants (both genders), that were short-, medium- and long term in our program. Diet was mainly composed of unprocessed vegetables/fruits, whole grains, legumes, potatoes, and nuts/seeds. LDL-cholesterol, triglycerides, systolic and diastolic BP were within targets for: 93%, 97%, 88% and 95% participants, respectively. In females (vs. males), total- and HDL-cholesterol were higher (mean): 3.8 (SD 0.7) vs. 3.4 (SD 0.9), *p* = 0.002 and 1.5 (SD 0.3) vs. 1.1 (SD 0.2) mmol/L, *p* < 0.001), systolic BP was lower (113 (SD 11) vs. 120 (SD 10) mmHg, *p* = 0.001), while there was no difference in diastolic BP (71 (SD 9) vs. 72 (SD 8) mmHg, *p* = 0.143). More females vs. males reached target triglycerides (99% vs. 91%, *p* = 0.021) and systolic BP (92% vs. 79%, *p* = 0.046), while similar females and males reached target LDL-cholesterol (94% vs. 91%, *p* = 0.500) and diastolic BP (93% vs. 100%, *p* = 0.107). Participation in our WFPB lifestyle program is associated with favourable dietary intakes, safety markers, and CV risk factor profiles.

## 1. Introduction

Over the past decade, adopting a strict plant-based diet (PBD) has become increasingly popular, with several dietetic organizations endorsing its benefits in terms of preserving cardiovascular health [1,2,3,4,5,6,7]. A number of observational studies and interventional trials comparing PBD with other types of diets have suggested that the healthiest eating habits might include a healthy version of PBD because of its potential to effectively sustain a healthy body weight and to prevent common non-communicable chronic diseases [8,9,10]. Conversely, concerns have been raised over the possible association of sub-optimally designed PBD with increased serum UA levels and gout development [11] as well as nutritional insufficiency [1,2]. Hence, recent research has focused on optimizing PBDs by addressing these potential shortcomings whilst preserving its effectiveness in terms of health maintenance. 

Several cross-sectional studies have assessed dietary intakes of adults on PBD [8,12,13,14,15,16]. Studies used various methods, from a food frequency questionnaire (FFQ) and three-day weighted dietary record (3-DR) to three 24-h DR and compared intakes either with control (vegetarian or omnivore) diet or with reference recommendations. Cross-sectional studies that used 3-DR and compared with dietary guidelines, for example on Swiss [14] and Finnish vegans [16], found certain nutrient insufficiencies for calcium as well as vitamins B_12_ and D in both studies. However, these studies were relatively small (53 and 22 participants, respectively) and did not include habitual dietary supplement intake in the final 3-DR analysis. This poses a limitation with respect to the estimation of the total dietary intake. In the Swiss study, participants were asked not to take supplements during the study duration, while in the Finnish study the type of dietary supplements used was recorded, but they were not included in the final analysis, which presents a limitation. 

A typical Western-type diet contains large amounts of refined sugar, added salt, dietary cholesterol, total and saturated fats, alcohol, and low amounts of whole grains, legumes, fruits, vegetables, and nuts. This imbalance plays a crucial role in the rising rates of type 2 diabetes, arterial hypertension, dyslipidaemia, obesity, and coronary artery disease [17,18]. Conversely, PBDs (vegetarian type of PBD included) have been shown to decrease the risk for morbidity and mortality [19,20] due to chronic non-communicable diseases, such as cardiovascular diseases [21,22], cancer [20], metabolic syndrome [23], type 2 diabetes [24,25,26,27], and obesity [28,29]. Furthermore, CVDs remain the leading cause of death globally [30], accounting for one-third of all deaths and an estimated 422 million prevalent cases in 2015 [31]. Dyslipidaemia [32] and high blood pressure (arterial hypertension) represent two leading risk factors for CVD [33]. Lifestyle intervention is a cornerstone in the prevention and management of CVD diseases, primarily addressing lifestyle behaviour factors, such as obesity, physical inactivity, tobacco and alcohol consumption, and stress. Hence, a healthy and optimal diet is one of the foundations of CVD risk reduction [34,35,36]. In this respect, PBDs interventions are effective in lowering plasma cholesterol [37] and blood pressure [38,39], but not triglyceride levels when compared to omnivorous diets [37].

Previously, we reported the effects of our WFPB lifestyle program on body-composition indices over 10 weeks [40] and CVD risk factors during 10–36 weeks [41]. There is still little known about the effects of short-, medium- and long-term strict PBD and lifestyle support on the dietary intake and CV health status of adults that were followed from the beginning of the PBD journey. In the present study, we investigated the influence of the short- (0.5–<2 years), medium- (2–<5 years), and long-term (5–10 years) WFPB lifestyle programs on dietary intake and CV heath status. We have tested three hypotheses (H_1_, H_2_ and H_3_). 

**Hypothesis 1** **(H1).**
*There is no significant difference in nutritional intake between the participants to our program in short-, medium- and long term (by gender).*


**Hypothesis 2** **(H2).***There is a difference in the lipid profile and BP status between the short- and medium-term, but not medium- and long-term groups (by gender). According to our clinical experiences and several randomized control trials that have utilized a WFPB diet* [42,43]*, we assumed that all the benefits for a lipid profile and BP status would be achieved within the first two years.*


**Hypothesis 3** **(H3).**
*At least 80% of all participants have plasma lipids and BP values within the recommended normal referenced values.*


## 2. Materials and Methods

### 2.1. Study Design and Eligibility 

This cross-sectional study took place in six Slovenian regions. We included free-living, heterogeneous participants who had been in our whole-food plant-based (WFPB) lifestyle program for 0.5–10 years. Based on the duration of the program, we divided participants into three groups: short-term (0.5–<2 years), medium-term (2–<5 years), and long-term (5–10 years). The study lasted from June to August 2019. It was conducted in accordance with the Declaration of Helsinki and the protocol was approved by the national Medical Ethics Committee of the Republic of Slovenia (approval document 0120-380/2019/17), as well as the Slovenian Ethical committee on the field of sport (approval document No. 05:2019). This trial was registered on 6 June 2019 at https://clinicaltrials.gov No. NCT03976479. After the participants were given a comprehensive explanation of the study, written informed consent was obtained from all participants.

### 2.2. Subject 

All participants had previously been on Western-type lifestyle and diet, they were not highly motivated, nor had any preference for PBD. The Western-type diet of the participants at baseline included foods of animal origin (milk and dairy products, meat and meat products, eggs and egg products, fish), refined grains and flour (i.e., white bread, pasta), sweets and pastry, animal fat and vegetable oils, sugar-sweetened beverages, as well as alcoholic beverages. It contained much less fresh vegetables and fruits, while unrefined and whole plant grains were largely absent. In short, this resulted in a high intake of fat, especially saturated fatty acids (SFA) and cholesterol, and a low intake of n-3 long chain polyunsaturated fatty acids (PUFAs). It also contained free sugars and more alcohol, while the fiber intake was much lower. Participants who joined our WFPB lifestyle program were at different points of the PBD continuum, from less consistent (i.e., plant-rich diet) to more consistent PBD (i.e., strict PBD, in our case supplemented WFPB diet with ≤3% of energy form animal protein). In this study, we included everyone meeting the inclusion criteria and who responded to our invitation through closed social media support groups or by personal contact with WFPB diet health coaches. The inclusion criteria for this study were: being on the supplemented WFPB diet (see Section 2.3.1) for 0.5–10 years. We did not set limits concerning dietary restrictions (e.g., gluten, tomatoes, peanuts, and citrus), current body mass index (BMI), or smoking. In this primary prevention setting, the exclusion criteria were: pregnancy or lactation, competitive or top-level athletes, major musculoskeletal restrictions, CVD, type 2 diabetes and use of medication affecting plasma lipids and glucose or blood pressure (recommended secondary prevention), active malignant, autoimmune, and neurodegenerative diseases), ≥3% of energy intake from animal protein, incomplete blood assay and unanswered questionnaires. A total of 370 participants met the inclusion criteria and were recruited after a two-stage interview process. Of those recruited, 44.8% agreed to participate (*n* = 166 participants signed the informed-consent form) while 55.2% declined to participate (*n* = 204) for one of the following reasons: distress associated with participation, challenging personal circumstances, vacation period and/or demanding methodology (time-consuming 3-DR), or discomfort against blood collection. Of the 166 participants, four discontinued their paritcipation, while we excluded 11 participants (six participants consumed >3% of energy from animal protein based on 3-DR evaluation, two had an incomplete blood assay, two had unanswered or uncompleted questionnaires, while one did not want to complete them). Thus, we included 151 adult participants (91% of initially included, Figure 1) aged 18–80 years in the final analysis. 

### 2.3. Intervention: WFPB Lifestyle Program

The WFPB lifestyle program comprised the nutritional part, physical activity (PA) component, and a support system (see Table 1). 

#### 2.3.1. Nutritional Part

The dietary pattern in our dietary program consisted of ≥90% of the energy from WFPB diet, defined by Campbell and Campbell (2005). The principles of the WFPB diet are: ad libitum intake of whole grains, fruits, vegetables, and legumes, moderate intake of nuts, seeds, avocados, soy (e.g., tofu) and wheat products, little or no added fats/oils (e.g., coconut, and palm fat/oil, olive oil), and the exclusion of all animal products. The WFPB diet is based on whole or minimally processed plant foods, while ultra-processed foods (defined by the NOVA classification system [44]), highly refined carbohydrates (white rice, white flour), foods with added sugars (table sugar, high-fructose corn syrup), and sweeteners are omitted [45]. In our program, the participants were advised to consume the majority of energy from starchy foods, such as whole grains, legumes, and potatoes, all prepared without oil or added fat. Participants were asked to limit portions of high fat plant foods (1–2 table spoons of flax, 1–2 table spoons of sesame seed/day, 20–30 g of walnuts, hazelnuts or almonds/day, occasionally pumpkin seeds (as part of salads, nut butters or smoothies), while minor amounts of soy products (e.g., tofu and soy beverage/soy milk) up to 2–4 times/week (mostly as ingredients). 

In our dietary program, we included ≤10% of daily energy intake from plant-based meal replacement (MR; 35–37 g soy or pea protein/100 g; 1–2 portions/day; 10–15 g of plant protein/portion) and dietary supplements (for all participants vitamin B_12_ and vitamin D_3_ (from October to May); optionally PUFA (eicosapentaenoic acid (EPA; C20:5*n*-3) and docosahexaenoic acid (DHA; C22:6 *n*-3) or others) to the WFPB diet, called the supplemented WFPB diet. We aimed to ensure nutrient adequacy, without adding excess energy [46], with MR that is safe, simple to prepare, effective, and convenient [47,48,49,50,51] and has long-term compliance [52]. The supplemented WFPB diet was individually optimized by an experienced health coach to meet nutritional needs, cultural preferences, and lifestyle. Despite the fact that weight management was a core part of our program, there was no need for calorie counting, and recommended WFPB diet foods were consumed ad libitum to full satiety at each meal [41]. 

#### 2.3.2. Physical Activity 

PA is incorporated in the WFPB lifestyle program as part of a healthy and active lifestyle. It includes habitual PA (part of daily life), as well as organized (guided or prescribed by health coaches, free of charge) and not-organized (part of healthy and active lifestyle). The participants were encouraged to engage in at least 2–3 weekly workouts for 45 min. Apart from this, the participants were also encouraged to perform at least 30 min per day of low- to moderate-intensity aerobic PA (i.e., brisk walking or biking) and the same, but for longer (1–2 h) of low- to moderate-intensity activity during the weekend. No unhealthy or extreme PA was encouraged. PA was promoted as an integrated part of healthy and active lifestyle, and not as a means for weight loss. 

#### 2.3.3. Support System

Furthermore, an integral part of the program was a support system including a lot of practical advice and a follow up (FU) that eased the transition from the Western-type dietary lifestyle to a WFPB lifestyle. It was conducted via different social media channels (mostly Facebook and Messenger closed groups and e-mails), workshops, and the FU. The goal of the social-media support groups was to provide simple and nutritious recipes (cooking instructions and hints pictures of delicious meals and short videos) to deliver accurate information about health and nutrition, to motivate and encourage participants to comply with the prescribed dietary regimen, to share experiences and cooking skills, and to help participants face daily challenges. FU was carefully integrated into other three components: body-composition measurements, personal consultation, and medical monitoring. Finally, regular medical monitoring was performed to control general health markers and markers of concern when adopting PBD (e.g., vitamin B_12_ and D (25OH), iron and others) three times at the baseline, approximately three months into the program and every year at later stages. No meditation or systematic stress techniques were introduced to the participants. 

### 2.4. Outcomes

#### 2.4.1. Dietary and Food Intakes

The dietary intakes from foods, MR, and supplements were assessed by 3-DR. The participants were given precise oral and written instructions on how to record 3-DR at home. We asked them to use our calibrated electronic kitchen scales in order to precisely weigh and record all foods and beverages consumed, excluding leftovers. In some cases, when exact weighing was not possible (e.g., in case of eating out), standard household measures (spoon, cup, glass, etc.) or a picture booklet (i.e., photos of reference foods with their actual mass in grams) [53] were allowed for semi-quantitative recording. We also instructed them to write the type, amount, and flavour of MR and dietary supplements consumed over three consecutive days (two weekdays and one weekend day). The study participants could choose the day of the beginning of dietary recording within a given period (i.e., Sunday, Monday, Tuesday or Thursday, Friday, Saturday). For the evaluation of dietary intake of the conventional WFPB diet, we used a dietary software, Open Platform for Clinical Nutrition (OPEN) [54], which is a web-based application, developed by the Jozef Stefan Institute [55] in Slovenia and supported by the Euro FIR AISBL [56] and the European Federation of the Association of Dietitians (EFAD). The dietary software has been upgraded to 3-DR methodology. The food intake data (from 3-DR) were used for the assessment of energy, macro- and micronutrients intakes. The energy and nutrient content of processed foods or homemade meals were estimated with recipe simulation using labelled ingredients and nutrient contents. Additionally, we asked participants to accurately record in what form the food (raw or cooked) was weighed and how they consumed it (e.g., raw, boiled in water, baked in the oven; examples: rice weighted raw, ingested cooked; carrots weighted raw, ingested cooked in the soup). We used appropriate conversion factors [57] when entering the dietary data into OPEN. For the purpose of this study, OPEN was continuously updated by adding specific plant-based dietary products. The dietary intake data were collected by investigators, further checked in detail, and evaluated by two experienced Master of Nutrition Engineering students. The precision of entries into OPEN were checked at least twice. In order to assess the nutritional intake from MR and dietary supplements, we used the services of Res-Pons, who professionally manage the database with all dietary supplements and medicine products in the Slovenian market [58]. We evaluated intakes from foods, MR, and dietary supplements separately, and are going to present them in detail in a separate manuscript. In this study, data about the intakes from foods, MR, and dietary supplements were combined in one database and presented as the dietary intake of energy, macro- and micronutrients, as well as food groups. Intakes of some main nutrients were compared to recommendations [59,60].

#### 2.4.2. Cardiovascular Risk and Safety Factors

For the biochemical assays, 10–15 mL of blood was taken after 10–12 h overnight fasted state. Lipid and other biochemical parameters were measured with standard laboratory tests in national medical biochemical laboratories with the same analytical methods. Total cholesterol (S-cholesterol), high-density lipoprotein (HDL-cholesterol), and triglyceride plasma levels were measured directly, and low-density lipoprotein cholesterol (LDL-cholesterol) levels were assessed using the Friedewald formula. For LDL–cholesterol, the intra- and inter-assay coefficients of variation ranges were 1.36%–2.26% and 2.34%–2.71%, respectively. Linearity was secured within a concentration range of 0.26%–10.3 mmol/L. The markers that we included in the blood analysis were serum S-cholesterol, LDL-cholesterol, HDL-cholesterol, triglycerides, uric acid (UA), and haemoglobin. Blood pressure was measured using the oscillometric technique in the supine position, after five minutes of rest. The average of two measurements three minutes apart were used for the analysis. Our blood and BP results were internally reviewed by a specialist of medical chemistry, the chair of the protein-lipid laboratory at the University Medical Centre in Ljubljana (see Acknowledgments).

The proportion of participants reaching the recommended guideline targets (The European Society of Cardiology (ESC) [61] and the European Atherosclerosis Society (EAS)) for LDL-cholesterol and triglycerides in the setting of primary CVD prevention was assessed [62].

### 2.5. Statistical Analysis

Statistical analysis was performed with R 3.5.2 with the dplyr [63], ggplot2 [64], and arsenal [65] packages. For numerical variables, we used ANOVA for the differences between different groups and Tukey’s post hoc test, where differences were statistically significant. Where the subsample was small, we referred to the Kruskal–Wallis test. We used the Dunn post-hoc test where the Kruskal–Wallis test was significant due to a smaller male sample. When we analysed dependent samples, we used a T-test for the dependent sample. Similarly, for the categorical variables, we used the chi-square test and Fisher’s exact test where the subsample was small. The threshold for statistical significance was 0.05.

## 3. Results

### 3.1. Characteristics of the Participants

We evaluated 3-DR from 154 participants, who also had completed blood samples and answered all questionnaires. From them, we excluded three participants (2%) as they consumed ≥3% of energy from animal protein (they consumed sea fish, as they were on vacation). In the final analysis, we included 151 participants (109 females (72%), 42 (28%) males) from six regions of Slovenia. The mean age and current BMI (mean (min–max)) were 39.6 (SD 12.5) years and 23.9 (17.7–41.4) kg/m^2^. Group 1 included 51 participants (35 females (69%), 16 males (31%)), group 2 included 56 (43 females (77%), 13 males (23%)), and group 3 included 44 participants (31 females (70%), 13 males (30%)). The average duration of the WFPB lifestyle program for all participants was 4.1 years (1.3 years for group 1, 3.9 years for group 2, and 7 years for group 3). All participants improved their baseline mean pre-obesity BMI range to a normal BMI range (from 26.4 to 23.9 kg/m^2^), and experienced decreased BM and BF % points (−7.1 kg and −6.4% points, respectively) (*p* < 0.001 for all). Participants were physically very active (Long International Physical Activity Questionnaire (L-IPAQ) score: 5542 METs min/week), had good sleep quality (Pittsburgh Sleep Quality Index (PSQI) score: 2.7), and perceived low stress (Perceived Stress Questionnaire (PSQ) score: 0.3). Detailed lifestyle and body composition status will be presented in a separate manuscript. None of the participants were concurrently on lipids, blood pressure, or blood sugar control medications, reflecting the primary prevention settings of the program (Appendix A). 

### 3.2. Dietary and Food Intakes of Participants in Our WFPB Lifestyle Program

There were no statistically significant differences among participants in the short-, medium- and long term in our WFPB lifestyle program by gender, in absolute intakes of energy, most macronutrients (except linoleic acid, α-linolenic acid for females and water for males), and a majority of micronutrients (see Table 2 and Table 3). Consequently, we present and comment upon dietary intakes for all participants, regardless of their time in the WFPB lifestyle program (from 0.5–10 years) by gender, (see Appendix A). 

The mean energy intake of all participants was 2057 (SD 689) kcal/day. The mean proportions of macronutrients for all participants, of both genders, were adequate with 20% of energy from fat, 57% from carbohydrates, 7% from fibres, and 15% from protein. The mean fibre intake of all participants was 70 (SD 21) g/day. The average total water intake, including drinking water, water from beverages, and solid foods was 4 (SD 1.2) litres/day. Participants had a very low intake of free sugars at 22 (SD16) g/day (4% of energy), saturated fats 7 (SD 4) g/day (3% of energy), and dietary cholesterol 7 (SD 18) mg/day (vs. recommended < 300 mL/day), with high intakes of linoleic acid and α-linolenic acid. Intakes of nutrients that are often of concern in PBD like protein (mean 1.2 (SD 0.45) g/kg BM), PUFA (EPA and DHA), vitamin B_12_, iron, and zinc (except for calcium: 1081 (SD 329) mg/day) were all adequate [59,60].

The mean energy intake was significantly lower in females as in males (1841 (SD 539) vs. 2618 (SD 726) kcal/day, *p* < 0.001). Consequently, the intakes of most macronutrients (nine out of 14) (exception: EPA, DHA, arachidonic acid, cholesterol, and water, which were not statistically significant different, Table 2) and micronutrients (15 out of 25) were lower in females than in males (Table 3). There were no differences in the relative intake of macronutrients (% of energy intake) between genders, except in the percentage of total sugar intake, which was higher in females vs. in males (17 (SD 4) vs. 16 (SD 4) %, *p* = 0.028). 

The comparison between genders according to different groups showed that, with the female participants, group 1 had a statistically significant lower relative intake of free sugars (% E) than group 3 (*p* = 0.032), while it showed a higher absolute intake of PUFA (*p* = 0.017) and linoleic acid (*p* = 0.041) compared to group 2, as well as a higher absolute intake of alpha-linoleic acid compared to group 2 (*p* = 0.050) and group 3 (*p* = 0.046). With the male participants, group 3 had a statistically significant higher intake of water compared to group 2 (*p* = 0.046) and group 1 (*p* = 0.041). The comparison between genders of the three groups did not show any significant difference when it came to the intake of micronutrients.

The evaluation of food group intakes showed that the supplemented WFPB diet was primarily based on unprocessed vegetables and fruits, whole grains, legumes, white potatoes, nuts and seeds, bread and bakery products, plant-based MR, spices and herbs and processed fruits (mean: from 455–20 g/day; Table 4). Intake of pasta was very low (mean: 17 (SD 35) g/day), mostly consumed in combination with PA. Intakes of fast food and ready meals, processed vegetables, sweet products, alcoholic drinks, vegetable fat, and of sweeteners were very low (mean: 6.5–0.2 g/day). Intakes of foods of animal origin were in minimal amounts (3–0.2 g/day for fish and meat; 0.1 g/day for milk and dairy products), while there was no consumption of eggs or added animal fat. The majority of vegetables (99% in females and males) and fruits (95% in females and 94% in males) were consumed fresh and unprocessed.

The majority of the group vegetables was represented by cruciferous (e.g., broccoli, kale, cabbage) and coloured vegetables. With regard to the fruits group, the most consumed fruits were berries, cherries, other local fruits (i.e., apples), as well as dried dates and bananas. Mostly consumed among grains and the products food group incorporated oatmeal, buckwheat porridge, and whole wheat bread. From the legumes group, beans, including green beans, lentils, chickpeas, and soybean tofu were the most consumed sub-groups. From the potatoes group, locally grown white potatoes were the most consumed, occasionally also sweet potatoes. From bread and bakery products, the majority of bread was wholegrain (from wheat, buckwheat and rye) and MR was consumed as part of breakfast and recovery after a resistance workout or as part of dinner. The nuts and seeds group mostly consisted of walnuts, flaxseeds, and unshelled sesame seed. Participants included all locally grown spices and herbs, in addition to bulbs, celery, rosemary, and turmeric. Processed fruits and vegetables were consumed as fresh homemade smoothies (green-fruits smoothies), while liquid intake came from water, herbal teas (e.g., green, black and hibiscus), fibre beverages, and hypotonic sport drinks. Salt intake was in most cases iodized and was used mostly for soups, burgers, spreads, and salads. 

All foods were prepared without added vegetables oils or fats of plant (coconut, palm) or animal origin (i.e., potatoes: boiled, mashed, or baked on baking paper). The majority of participants did not use any vegetable oils, not even extra virgin olive oil, while only a few individuals used minor amounts for salads dressing (mean vegetable oil and fat intake = 1.4 g/day). The majority of sugar was consumed as naturally occurring sugar, from dates and other fruits, while free sugars originated from MR, smoothies and hypotonic sport drinks. Alcohol was not part of the participants daily diet since only three individuals were drinking alcohol (two participants 1 dcl of wine and 4 dcl of beer within three days, one participant 1 dcl of wine on the first and third days of 3-DR). 

The majority of energy from non-conventional foods came from MR in powder (mixed in water or in plant beverage without flavour, oil or sweeteners) and hypocaloric sport drinks, while a very limited amount came from dietary fibre beverages and herbal teas. The majority of micronutrients from non-conventional foods came from MR, multivitamin dietary supplements, and selected single dietary supplements (e.g., vitamin B_12_ and D_3_). There was no specific pattern in the selection of dietary supplements brand, since the participants used supplements (e.g., vitamin B_12_, D_3_, probiotics) from approximately 40 different producers. 

### 3.3. Cardiovascular Health and Safety Factors Status of Participants in Our WFPB Lifestyle Program 

#### 3.3.1. Main Findings

Cardiovascular health and safety factors status are shown in Table 5 and Appendix A. Lipids and BP status were our main study endpoint in terms of cardiovascular health. There were no statistically significant differences among participants short-, medium-, and long term (by gender) in our WFPB lifestyle program, in lipids, or BP status, with the exception of LDL-cholesterol in females, being lower in those that were the longest in our program (mean, mmol/L; group 1, 2 and 3: 2.2 (SD 0.6); 2.1 (SD 0.7) and 1.8 (SD 0.5) mmol/L, *p* = 0.025). 

Due to this, we comment below on the lipids and BP status for all participants together, by gender, regardless of their time in the WFPB lifestyle program (see Appendix A). Females had significantly higher total cholesterol (3.8 (SD 0.8) vs. 3.4 (SD 0.9) mmol/L, *p* = 0.002) and HDL-cholesterol (1.5 (SD 0.3) vs. 1.1 (SD 0.2) mmol/L, *p* < 0.001), while lower triglycerides (0.8 (SD 0.3) vs. 1.0 (SD 0.4) mmol/L, *p* = 0.037) and systolic BP (113 (SD 11) vs. 120 (SD 10) mmHg, *p* = 0.001) than males. Laboratory variables included two additional safety markers, serum UA and haemoglobin. For all participants the average serum UA concentration was 272 (SD 68) µmol/L. Females had significantly lower serum UA (245 (SD 50) vs. 346 (SD 52) µmol/L, respectively, *p* < 0.001) and haemoglobin (137 (SD 10) vs. 153 (SD 9 g/L, *p* < 0.001) than males. 

In terms of cardiovascular health, we have tested two hypotheses (H_2_ and H_3_). Comparing the difference in lipid profile and BP status between the short-, medium-term, and long-term groups by gender, we cannot fully confirm the hypothesis H_2_. Contrary to the set hypothesis, females in group 3 had statistically significant lower LDL-cholesterol compared to the group 1 (*p* = 0.034). Other differences between groups 2 and 3 by gender were not established. Hypothesis H_3_ stated that at least 80% of the sample has plasma lipids values and BP within the recommended target values. The European Society of Cardiology (ESC) and the European Atherosclerosis Society (EAS) [61,62] do not establish target values for total or HDL-cholesterol, but recommend LDL-cholesterol levels below 3 mmol/L and triglycerides levels below 1.7 mmol/L in the primary prevention of CVD. For the total sample, 140 (93%) participants had LDL-cholesterol levels and 146 (97%) participants had triglyceride levels within target values. Significantly more females achieved target triglyceride levels (108 (99%) vs. 38 (91%), respectively, *p* = 0.021). For BP, we have used the upper normal values of ≤ 129/84 mmHg [61]. BP comparison with referenced cut off values showed that 133 (88.1%) and 143 (94.7%) participants had systolic and diastolic BP, respectively, within range. However, more females than males achieved target systolic BP (100 (92%) vs. 33 (79%), respectively, *p* = 0.046), but not diastolic BP (42 (100%) vs. 101 (93%), *p* = 0.107). The hypothesis H_2_ was therefore fully confirmed for the whole sample, but only partially when split by gender. Hypothesis H_3_ stated that there is difference in lipid profile status between the group 1 and 2, but not between the group 2 and 3. When analysed a group comparison we cannot fully confirm the hypothesis H_3_. The female group 3, in contrast to the set hypothesis, had significantly lower LDL-cholesterol compared to group 2. Other differences between groups 2 and 3 were not established.

#### 3.3.2. Safety Markers

We have included two safety markers from the laboratory variables that we considered critical when adopting PBD, namely serum UA and haemoglobin concentration (Table 6). There is no clear consensus about the normal serum UA reference range, however, according to Italian researchers, several European countries have identified serum UA values as normal between 308–428 mmol/L for adult males and postmenopausal females and between 154–357 mmol/L for premenopausal females [67]. Compared with the proposed reference, 132 participants (87.4%) had normal values of serum UA with a significant difference between genders (106 females (97%) and 26 males (62%), *p* < 0.001). Five participants (3.3%) had an increased value (1 female (1%) and 4 males (9%), while 14 participants had too low values (2 females (2%) and 12 males (29%). Furthermore, according to the World Health Organization (WHO), the recommended haemoglobin cut-off level for non-pregnant females (≥15 years) and males (≥15 years) is ≥120 g/L and ≥130 g/L [68]. Our study showed that 146 participants (96.7%) had recommended haemoglobin concentration values with the highest proportion of normal in group 3 (100%) and with a significant difference between the genders (*p* < 0.001).

## 4. Discussion

### 4.1. Main Findings

Our results indicate that the proposed WFPB lifestyle is associated with a favourable dietary intake and cardiovascular risk profile in apparently healthy adults (primary prevention setting). The evaluation of dietary intake of participants showed that the WFPB lifestyle program provides a similar dietary intake over time in both genders. We partly confirmed hypothesis H_1_ that there is no statistically significant differences among participants in the short-, medium-, and long term in our WFPB lifestyle program by gender, in absolute intakes of energy, most macronutrients (except PUFA, linoleic acid, α-linolenic acid for females and water for males), and all micronutrients. We have also confirmed hypothesis H_2_ for the overall sample (but not when considering only the male group; 79% instead of ≥80% to be exact) which stated that at least 80% of the total sample had plasma lipid values and BP within the recommended normal referenced values that indicates low CVD risk. However, we did not fully confirm hypothesis H_3_, which proposed that there would be a significant differences in lipid profiles between groups 1 and 2, but not between groups 2 and 3. In terms of the further improvement of lipid profile results for LDL-cholesterol for the third female group, we can attribute this partly to their lower maximal lifetime BM, baseline BM (will be presented in a separate manuscript), and current BMI. We suspect that this statistically significant difference might also be associated with the longer time on supplemented WFPB diet, or it might be due to non-significant, but meaningful differences in PA levels (mean, total METs min/week; group 1, 2 and 3: 5265 (SD 3579), 4483 (SD 3299) and 6438 (SD 4320), *p* = 0.086).

### 4.2. Dietary Intake and Diet Quality

It is important for cardiovascular health that WFPB lifestyle is not just effective, but nutritionally adequate, providing adequate intake of energy and nutrients, while limiting the intake of unfavourable nutrients (e.g., free sugars, salt, trans and saturated fats, dietary cholesterol), and achieving sustainability in terms of adherence. We found a lower intake of calcium for females than recommended (84% of recommended) [59], lower intake of vitamin D for both genders than recommended, and a higher intake of vitamin B_12_ for both genders. The reason for the low intake of vitamin D was that participants were advised to interrupt their intake of vitamin D_3_ supplements from May to October (summer time), since the UV index for Slovenian latitude provides adequate sun exposure to synthesise vitamin D_3_ in the skin. The mean intake of vitamin B_12_ was 272 (SD 393) µg/day which is about ten times higher than recommended (20 µg/day for all age groups) [59]. As the absorption rate of vitamin B_12_ taken as dietary supplements is only 1%, the relative increase in serum vitamin B_12_ is related to the log of the dose [69]. In our study, the majority of participants used vitamin B_12_ in methyl cobalamin form, 1000 µg per serving, two times per week (those aged <50 years) and three times per week (those aged >50 years). 

A comparison with six cross-sectional studies (Table 7) showed that participants in our study consumed a comparable amount of energy (one theoretical study on WFPB diet is for optimal referenced intakes) and a lower amount of total fat and saturated fat, and more fibre, vitamin B_12_, and selected minerals except for sodium.

In our opinion, the main reasons for the achieved very low serum cholesterol plasma values of participants in our study are low intakes of saturated fat and total fat, and a high intake of dietary fibre ((mean, g/day) of 64 (SD 16) and 85 (SD 45) g/day, in females and males, *p* < 0.001). A high intake of dietary fibres and whole grains is a very important aspect of PBD. A systematic review of 185 prospective studies and 58 clinical trials, showed striking dose-response evidence of a high intake of dietary fibres and whole grains for reduced incidence and mortality from several non-communicable diseases, including CVD. Analysed prospective studies showed even more striking reductions in, and dose-response relationships between all-cause mortality, total cancer deaths, and total CVD deaths and incidence, stroke incidence, and incidence of colorectal, breast, and oesophageal cancer [70].

**Table 7 nutrients-12-00055-t007:** Dietary intake comparison with different cross-sectional studies on PB dieters.

Country	Belgium [8]	Switzerland [14]	Finland [16]	France [66]	U.S. [71]	U.S. [72]	Slovenia (Our Study)
Year of publication	2014	2015	2016	2017	2018	2019	2019
Participants (*n*)	*n* = 104	*n* = 53	*n* = 22	*n* = 789	Theoretical ^†^	*n* = 200/Theoretical	*n* = 151
Age (years old)	20–69	18–50	24–50	>18	-	-	18–78
Diet name	Vegan diet	Vegan diet	Vegan diet	Vegan diet	Vegan diet	WFPB diet	supplemented WFPB diet
Animal food limit	Excluded	-	Self-defined	Excluded	Excluded	Excluded	≤3% of energy as animal protein
Diet assessment	FFQ *	3-DR	3-DR	3 × 24–h DR	7 × single day	30-day meal plan	3-DR
Duration on PBD (y, min–max)	-	3.0 (1.0–18)	8.6 (2–16)	Currently	-	-	4.1 (0.8–10.3)
Dietary intake/day							
Energy (kcal)	2383	2469	2150	Adjusted	1302	Adjusted	2058
Total fat (g)	68	96	88	72.7	35	38	44
Total fat (% E)	25	33	36.5	35.2	24	17	20
SFA (g)	21	20	21	19.4	6	6.5	7
SFA (% E)	8	7 ^‡^	8.6	9 ^‡^	4 ^‡^	3	3
Cholesterol (mg)	149	12	44	55.4	0	0	7
Carbohydrates (g)	336	324	252	235.7	225	365	287
Carbohydrates (% E)	57	54	49	51.2	69	73	57 (64% E)
Sugar (g)	156	180	-	105.3	109	-	85
Dietary fiber (g)	41	52	41	34.1	49	70	70 (7% E)
Proteins (g)	82	65	74	66.6	51	81	77
Proteins (% E)	14	11	13.7	13.2	15.7	16	15
B_12_ (µg)	-	0.2	0.9	2.7	0.7	904	280
Sodium (mg)	1316	2994	-	2589.6	-	2807	2075
Calcium (mg)	738	817	1004	760	782	959	1099
Iron (mg)	23	22.9	21	18.6	17.2	26	36
Zn (mg)	-	11.5	12	10	8.3	-	20

* FFQ is food frequency questionnaire. ^†^ Weight loss meal plan. ^‡^ Calculated from SFA and energy intake.

The quality of our supplemented WFPB diet of participants in this study can also be observed from the food groups’ intakes. Females and males consumed approx. 1200 g and 1300 g/day of foods from five food groups (e.g., vegetables, fruits, grains, legumes, and potatoes), and 80 and 100 g/day of (nuts, seed, spices and herbs) per day. At the same time, they consumed minimum amounts of highly processed foods, refined carbohydrates, oils and fats, sweetened beverages, alcoholic drinks, as well as foods of animal origin. A comparison with two other European cross-sectional studies showed that participants in our study consumed more vegetables, fruits, legumes, and potatoes, and less vegetable fat. In Table 4, only two cross-sectional European studies were suitable to include for comparison.

### 4.3. Cardiovascular Risk Factors 

Our results show that the supplemented WFPB diet is associated with favourable blood lipid and BP profiles. Blood lipid and BP levels were within recommended target ranges across all groups, suggesting both short-term and long-lasting effectiveness of the WFBP lifestyle. Our results extend favourable lipid results from our previous 36-week long intervention [41] with the total cholesterol of a whole sample of 3.7 (SD 0.8) mmol/L, LDL-cholesterol 2.0 (SD 0.7) mmol/L, HDL-cholesterol 1.4 (SD 0.4) mmol/L, triglycerides 0.9 (SD 0.4) mmol/L, and systolic and diastolic BP 115 (SD 11) and 71 (SD 9) mmHg.

Furthermore, females had significantly higher total- and HDL-cholesterol, with lower triglycerides and systolic BP than males (see Appendix A). The female group also reflected gender-specific differences in HDL-levels (1.5 (SD 0.3) mmol/L vs. 1.1 (SD 0.2) mmol/L for males, *p* < 0.001). 

Females in group 3 had statistically significant, lower LDL-cholesterol compared to group 1. Other differences between the gender groups were not established. 

In terms of safety markers, serum hemoglobin levels in both males and females, and serum UA levels in females were within ranges in the vast majority (>90%) of participants. In one third of males, however, serum UA levels were off the recommended range—predominantly due to levels below the recommended threshold of 308 mmol/L [67]. While elevated serum UA levels have been linked to gout, kidney stones, cardio metabolic risk and mortality, a U-shaped association between serum UA levels and mortality has recently been suggested, with very low serum UA levels (i.e., below 200 mmol/L) possibly associated with a significant—albeit moderate—increase in the risk of all-cause mortality [73]. Thus, while still relatively small, a proportion of male participants in our WFPB lifestyle program displayed a tendency towards low serum UA levels, which calls for further research into the possible association between PBDs, serum UA levels, and clinical events, especially in males.

Both blood lipid levels and BP in our study neatly fit within recommended ranges by the European Society of Cardiology (ESC) and the European Atherosclerosis Society (EAS) [61,62]. This may be somewhat expected, as the basic notions of the WFBP lifestyle are consistent with the evidence-based recommendations put forward by the ESC and EAS for CVD prevention (Table 7)—namely (i) no exposure to tobacco, (ii) a healthy diet low in saturated fat with focus on wholegrains, vegetables, fruits (without fish but with EPA and DHA), (iii) at least 150 min of moderate or 75 min of vigorous PA/week (or a respective combination thereof), and (iv) maintaining healthy weight (BMI between 20–25 kg/m^2^), a BP below 140/90 mmHg, LDL below 3 mmol/L and triglycerides below 1.7 mmol/L [62]. Furthermore, participants of the WFPB lifestyle program also adhered to specific lifestyle changes and functional foods recommended by the ESC and EAS, which are purportedly associated with improvements in lipid profile, namely, avoiding trans fats and alcohol (practically no alcohol intake by our participants), reducing saturated fats intake and BM, accompanied by an increasing intake of n-3 polyunsaturated fats and dietary fibre, use of functional foods enriched with phytosterols, and habitual PA [62].

Dietary intervention remains a cornerstone of management of dyslipidaemia and arterial hypertension, because it is safe and cost-effective in terms of CVD prevention [34,35,36]. Our study suggests the supplemented WFPB diet achieves target lipid levels, which is consistent with previous scientific reports. A systematic review and meta-analysis of 11 randomized controlled trials (including seven trials in vegans and four in lacto-vegetarians) provides evidence that vegetarian diets effectively reduce total cholesterol, LDL-cholesterol, HDL-cholesterol, and non–HDL cholesterol levels [74]. In a recent meta-analysis of 46 observational studies comparing the effects of vegan and omnivorous diets on cardio-metabolic factors, the mean LDL-cholesterol in vegans was 2.36 mmol/L (ranging from 1.75 to 3.31 mmol/L), mean triglycerides were 1.1 mmol/L (between 0.56 and 1.91 mmol/L), and mean blood pressure was 118/77 mmHg [75]. In a less representative study of 21 raw vegans, Fontana et al. found CVD risk factor status to be even more impressive: total cholesterol was 3.7 mmol/L, LDL-cholesterol 1.69 mmol/L, HDL-cholesterol 1.45 mmol/L, triglycerides 0.63 mmol/L, and blood pressure 104/62 mmHg [76]. Brazilian researchers [77] in their small study on 18 vegan individuals of both genders were 3.6 mmol/L, LDL-cholesterol 1.8 mmol/L, triglycerides 0.9 mmol/L, and HDL-cholesterol 1.4 mmol/L. One of the most comprehensive nutritional studies ever considered dietary, lifestyle, and disease characteristics of 6500 participants that were selected from 65 counties and 130 villages in rural China. The mean serum total cholesterol was 3.3 mmol/L. The diet, assessed by 3-DR, showed that their energy intake per kg of BM was about 30% higher in China than in the U.S., while the prevalence of obesity was much lower in China. The Chinese diet was composed as follows (mean daily energy intake): 14% fat, 71% carbohydrates (33 g fibers/day) and 15% protein. Animal protein intake was less than 1% of total energy intake (11% of energy from protein). During the study time, the U.S. adults had 16.7 times higher coronary disease mortality in males and 5.6 times higher in females compared to Chinese males and females. The researchers’ major finding concerning coronary artery disease was that the risk decreases with an increased consumption of plant-based foods and decreased consumption of animal-based foods [78]. Many individuals on a well-designed strict PBD might achieve very low LDL-cholesterol. A growing body of evidence depicts a log-linear relationship between LDL-cholesterol and CVD risk, suggesting the lower LDL-cholesterol, the better cardiovascular health, without a discernible lower LDL-cholesterol limit, below which the risk of morbidity might start to increase [62]. Conversely, in terms of HDL-cholesterol levels, the evidence is less conclusive. While seminal observational studies have shown an inverse relationship between HDL-cholesterol and atherosclerotic risk, recent evidence suggest that high [62] and extremely high HDL-cholesterol [79] may not reduce the risk of CVD events or may even be associated with increased all-cause mortality, thus suggesting a U-shaped relationship between HDL and CVD risk. Notwithstanding, HDL-cholesterol levels of our study population were within the “optimal” 1–2 mmol/L range, which is associated with the lowest mortality risk [79]. Along with low triglyceride levels, moderately high HDL-levels also suggest a favourable metabolic lipid profile of our study participants. Meanwhile, dysmetabolic dyslipidaemia (high triglycerides/low HDL) is associated with atherosclerotic events and is principally driven by metabolic derangements secondary to obesity, the metabolic syndrome, and/or insulin resistance, and adherence to the WFPB lifestyle seems to contract these mechanisms to yield low triglycerides, moderately high HDL, and low blood pressure. 

Arterial hypertension represents another major risk factor for CVD [33], and our results suggest that adherence to the WFBP lifestyle may provide a suitable option for the reduction of blood pressure. A recent review put forward several possible mechanisms by which PBD may achieve blood pressure control, namely vasodilation, antioxidant and anti-inflammatory effects, improved insulin sensitivity, decreased blood viscosity, altered baroreceptor function, modifications in the renin-angiotensin and sympathetic nervous systems, and alterations of gut microbiota [80]. Moreover, a recent meta-analysis of 11 RCTs on the effect of strict PBD on blood pressure in adults found comparable efficacy of a strict PBD without caloric restrictions, dietary approaches recommended by medical societies, and portion-controlled diets [39]. Of note, another recent RCT found that unmonitored lifestyle modification through diet and exercise, but not diet alone, was effective for lowering BP [79]. 

### 4.4. Support System

The essence of the support system is to optimally determine the support individuals or groups to follow the proposed lifestyle change and to maintain a long-term healthy and active lifestyle as the “new normal”. The support system is essential for adherence to lifestyle change. In our practice of more than two decades, we have tried and failed, but readjusted the model. Finally, we have found at least three key factors which currently enable successful change, but work only in combination, with (1) intellectual information, (2) an extensive support system and (3) increased socialization. Apart from the services, which we mentioned in the Method section, people adopt healthier behaviour if lifestyle coaches treat their goal personally and take on their challenge wholeheartedly. The entire toolbox of support systems for a long-term dietary and lifestyle change needs to become the family lifestyle, both because of the necessary support of the family, convenience of preparing meals and the notion of thinking in the same direction about a healthy and active lifestyle. Furthermore, an individual’s goals need to be related to the quality of life, to become a role model for the family, friends, and colleagues and to help others with first-hand experiences and sharing information about personal transformation [81]. Several predominantly PBD or strict PBD studies have confirmed the importance of the support system and success associated with it [17,78,82,83,84]. However, in order to implement the strict PBD lifestyle, there are several key issues for participants to address, i.e., (1) nutritional adequacy (e.g., protein, vitamin B_12_, vitamin D, iron, calcium and others), (2) possibility of increased flatulence when consuming increased quantities of legumes, (3) financial aspect of buying PB foods, which tend to be more expensive, and (4) reluctance to consume foods thought to be unpalatable, and others which needs to be communicated to increase the client’s/patient’s engagement [85]. In conclusion, the support system might not prove as effective if their constituent components are not associated with a respectful relationship and health-oriented social community to facilitate and maintain new lifestyle behaviour.

## 5. Strengths and Limitation

Performing a cross-sectional, nationwide multiregional study, with participants known to the health coaches promoting a plant-rich and WFPB lifestyle, allowed us to include all volunteers, supplemented WFPB dieters without BMI limitations, while knowing their maximal BMI, baseline BMI, and BF percentage (details will be provided in other manuscript) and who were eligible according to inclusion/exclusion criteria. Another strength was that, to the best of our knowledge, this was the first study which documented the long-term effects of a PBD lifestyle on common CVD risk factors in healthy and active adults whom we followed through their entire transformation from a Western-type lifestyle, while providing them with a constant extensive WFPB lifestyle support system. The participants were not highly motivated before entering the WFPB lifestyle, nor had they any inclination toward PBD which was evident from their maximal BMI and baseline body composition status, and their background was mostly middle class. Using 3-DR was a unique strength of the study, since our participants were a part of an ongoing and extensive support system. By using 3-DR it was possible to precisely assess participants’ actual dietary intake, including conventional foods as well as MR and all dietary supplements. We also used a very advanced database enabling us to calculate the intake of free sugars, which are rarely reported in other studies. The proposed paradigm could be implemented among the general population, since the study had a relatively dispersed geographical representation and participants’ regions and living environment types (urban, suburban, or rural) were documented. Furthermore, blood assays were measured with standardized methods and in national medical centres, and additionally reviewed by two experts, including a specialist of medical chemistry and a cardiologist (the co-author), both affiliated with the University Medical Centre Ljubljana. Strengths of the study also include a very high proportion of participants who completed the study within inclusion/exclusion criteria (91% of those that signed the informed consent form). An additional strength was that all participants were recruited within 30 days from the study registration, and the study finished in less than two months. 

Our study has some obvious limitations inherent to studies of heterogeneous multicity free-living subjects and cross-sectional nature of the investigation. As a limitation, we see a constant need for a larger cross-sectional sample (especially of males) and for performing a non-randomized study. We assumed that we might have lost the power of the male sample, since it is known from WFPB dietary-intervention studies that the ratio of women to men in the sample is strongly in favour of women. In addition, we cannot exclude the possible unknown impact of people who were within the set criteria, but did not respond or were not willing to participate in the study, so the results of this analysis are therefore applicable to those participants who attended the long-term assessment and are not generalizable to others that had not adopted a WFPB lifestyle. Another limitation is also not having collected participants’ baseline CVD risk markers and the limits which normally come from the single-time assessment with 3-DR. An important limitation related to CVD risk markers is also not measuring fasting glucose, which is a significant CVD risk factor. Additionally, our results were not limited to diet only, since the participants started a healthy and active lifestyle when entering into a supplemented WFPB diet lifestyle. Further, regular PA is known to be associated with lower BMI and BF percentage [86], especially in combination with a weight-management or dietary program [87]. Low and moderate intensity PA is also associated with a reduction of total, but not LDL-cholesterol [88]. Additional stimulus for better success of the supplemented WFPB diet was the extensive support system [41], since behavioural change [89] and motives related to personal well-being and heath [90] may represent a crucial driver to remain a long-term plant-based dieter. It would be valuable to know the dietary intake and cardiovascular health status of participants in our program who decide to follow a less consistent PB dietary lifestyle than the strict PBD.

## 6. Conclusions

Our results suggest that short-, medium-, and long-term WFPB lifestyles may be associated with long-term, sustainable, and favourable CV health results for those who have followed the program. There were no statistically significant differences between the intakes of energy and most nutrients among females and males, that were short-, medium-, and long term in our WFPB lifestyle program. There were no statistically significant differences among participants that were short-, medium-, and long term (by gender) in our WFPB lifestyle program, in lipids or BP status, with the exception of LDL-cholesterol in females, being lower in those that were the longest in our program. Further, 93% participants had LDL-cholesterol and 97% participants had triglyceride levels within target values. Significantly more females achieved target triglyceride levels, and 88% and 95% of participants had systolic and diastolic BPs within the reference range. Significantly more females than males achieved target systolic BP values. The proposed WFPB diet lifestyle program showed success also in maintaining safety biomarkers of haemoglobin and serum UA. It would be valuable to have a long-term prospective RCT with a high compliance rate to verify if we could duplicate our results in the wider general population.

## Figures and Tables

**Figure 1 nutrients-12-00055-f001:**
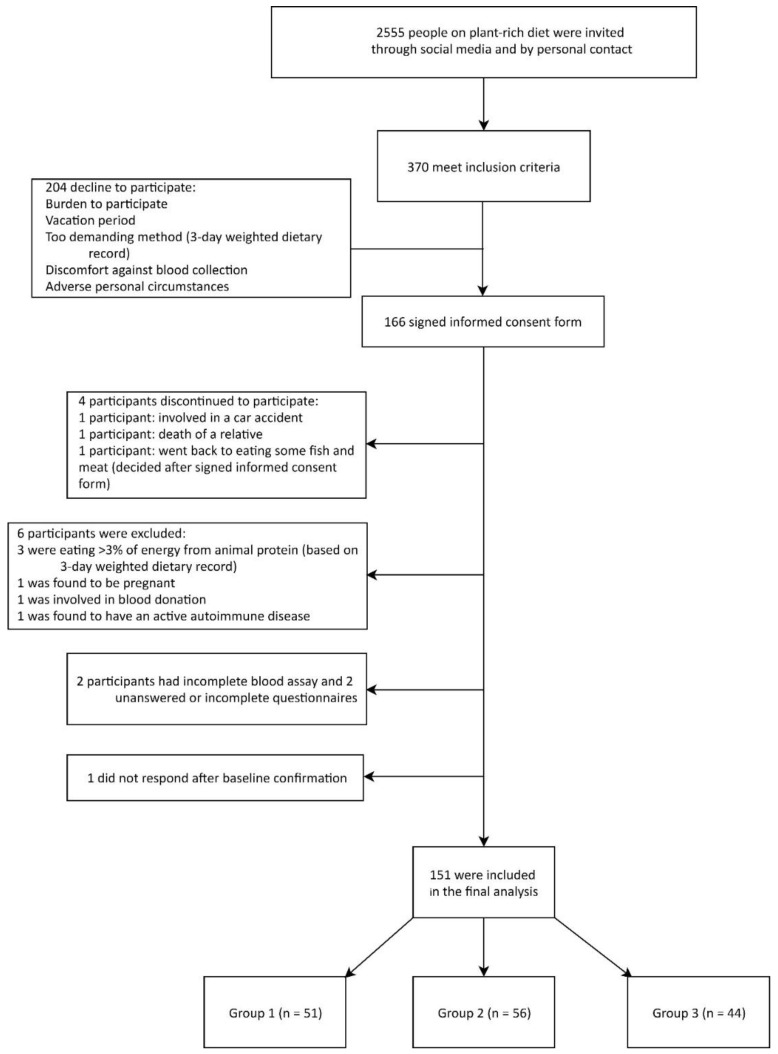
Enrolment of the participants and completion of the study. 3-DR: 3-day weighted dietary record.

**Table 1 nutrients-12-00055-t001:** A whole-food plant-based (WFPB) lifestyle program.

Nutrition	Physical Activity (PA)	Support System
Supplemented WFPB diet	Healthy and active lifestyle	Included a social-media component
WFPB diet ≥90% of the energy intake	Habitual PA (part of daily life) Organized (guided or prescribed, free) Other PA	Facebook Messenger e- mails
^PB^ MR ≤10% of the energy intake	Resistance-exercise activity (own body mass or fitness): 45 min: 2–3 times/week Brisk walking: 30 min/day Brisk walking or hiking: 1–2 h/weekend	Meal recipes (process, pictures, video) Health and nutrition topics Motivation Discussion board
Dietary Supplements ^†^ Vitamins: B_12_, D_3_, EPA and DHA *n*-3	Grouped Individual Fit challenges	Practical advice Meal plan evaluation (daily or weekly) Grocery shopping tour (three supermarkets) Cooking class (three hours)
Individually optimised meal plan Adjusted after 4–6 weeks: Details on food preparation, combinations, raw/cooked ratio	Health General fitnessBody composition	Follow-up Body-composition measurements Personal consultations Regular medical monitoring (at baseline, 3rd month, yearly)
60-min lecture/week for 10 weeks	No unhealthy or extreme PA	Constant and dynamic improvement
Healthful, ad libitum, tasteful, affordable	Independence of participants	No meditation or systematic stress techniques introduced

^PB^ Plant-based meal replacement (MR). ^†^ Vitamins: B_12_: whole year: 1000 µg/day, 2–3-times/week, D_3_ (October–April: 3000 IU/day; 7-times/week), *n*-3 long-chain polyunsaturated fatty acids: EPA and DHA (from fish oils; whole year): 625 mg/day.

**Table 2 nutrients-12-00055-t002:** Intake of energy and macronutrients by gender and to the time in WFPB lifestyle program: short (0.5–<2 years), medium (2–<5 years) and long-term (5–10 years) (groups (1, 2 and 3)).

	Group 1 (*n* = 51)		Group 2 (*n* = 56)		Group 3 (*n* = 44)		*p*-Value
Macronutrients (Per Day)	Female	Male	Female	Male	Female	Male	F/M
Energy intake (kcal)	1963 (501)	2683 (2069, 2968)	1734 (529)	2502 (2195, 2971)	1851 (577)	2832 (2123, 3056)	0.174/0.900
Carbohydrates (g)	284 (80)	366 (280, 412)	250 (85)	389 (312, 448)	270 (96)	364 (310, 460)	0.222/0.517
(% E)	58 (6)	53 (50, 59)	57 (4)	58 (55, 62)	58 (6)	58 (53, 65)	0.685/0.115
Total sugars (g)	86 (32)	96 (76, 119)	74 (23)	119 (83, 130)	84 (48)	90 (65, 98)	0.233/0.305
(% E)	18 (5)	16 (13, 18)	17 (4)	17 (15, 19)	18 (5)	13 (10, 17)	0.963/0.095
Free sugars (g)	19 (18)	21 (15, 27)	21 (11)	29 (19, 36)	26 (24)	20 (17, 28)	0.304/0.197
(% E)	4 (3)	3 (2, 4)	5 (2)	4 (3, 5)	5 (3)	3 (3, 4)	**0.042**/0.172
Starches (g)	90 (45)	129 (78, 164)	78 (47)	119 (86, 154)	79 (42)	145 (108, 165)	0.439/0.680
(% E)	18 (7)	19 (15, 22)	17 (7)	19 (15, 20)	17 (6)	19 (18, 28)	0.698/0.611
Dietary fibre (g)	68 (16)	90 (69, 102)	61 (16)	80 (71, 90)	63 (17)	80 (72, 96)	0.144/0.665
(% E)	7 (1)	7 (6, 7)	7 (1)	6 (5, 7)	7 (1)	7 (6, 7)	0.237/0.254
Fat (g)	44 (18)	67 (47, 86)	38 (13)	49 (37, 64)	40 (17)	57 (44, 81)	0.305/0.401
(% E)	20 (5)	24 (19, 28)	20 (4)	19 (13, 21)	19 (5)	20 (15, 25)	0.695/0.155
SFA (g)	7 (2)	11 (8, 14)	6 (3)	8 (6, 11)	6 (2)	10 (7, 14)	0.849/0.356
(% E)	3 (1)	3 (3, 4)	3 (1)	3 (3, 3)	3 (1)	3 (3, 4)	0.530/0.340
MUFA (g)	12 (6)	17 (14, 22)	11 (4)	13 (10, 21)	10 (5)	18 (15, 18)	0.317/0.559
(% E)	5 (2)	7 (5, 8)	6 (2)	4 (4, 7)	5 (2)	5 (5, 7)	0.094/0.404
PUFA (g)	20 (8)	29 (21, 45)	16 (6)	24 (15, 28)	17 (7)	23 (16, 37)	**0.017**/0.358
(% E)	9 (3)	11 (8, 13)	8 (2)	7 (6, 9)	8 (2)	8 (6, 11)	0.067/**0.047** ^a,b^
Linoleic acid (g)	13 (7)	21 (11, 31)	10 (5)	10 (10, 20)	11 (5)	16 (11, 29)	**0.050**/0.430
Alpha-linolenic acid (g)	6 (3)	7 (5, 13)	4 (2)	5 (4, 6)	4 (3)	6 (3, 8)	**0.026**/0.084
EPA (mg)	333 (284)	375 (104, 377)	350 (176)	378 (250, 379)	307 (242)	375 (253, 377)	0.739/0.436
DHA (mg)	235 (209)	250 (70, 250)	235 (119)	250 (250, 459)	205 (162)	250 (172, 250)	0.696/0.531
ARA (mg)	2 (7)	0.1 (0, 2)	2 (6)	0 (0, 0.4)	2 (5)	0.1 (0, 7)	0.980/0.156
Cholesterol (mg)	8 (24)	0.01 (0, 5)	4 (8)	0 (0, 20)	7 (20)	0 (0, 0.3)	0.657/0.971
Protein (g)	73 (20)	96 (82, 114)	67 (18)	98 (91, 115)	72 (18)	111 (82, 118)	0.330/0.903
(% E)	15 (2)	14 (14, 16)	16 (2)	16 (15, 17)	16 (2)	15 (14, 16)	0.087/0.308
(g/kg body mass)	1.1 (0.4)	1.3 (1.1, 1.5)	1.1 (0.3)	1.3 (1, 1.3)	1.2 (0.3)	1.2 (0.9, 1.5)	0.1300.916
Alcohol (mL/day)	0.3 (1.0)	0	0	0	0.2 (1.2)	0	0.323/0.973
Water ^w^ (L)	3.8 (1.1)	3.6 (3.2, 4)	4.0 (1.2)	4.5 (3.8, 5.7)	4.0 (1.3)	4.4 (4.1, 5.7)	0.652/**0.024** ^a^

The female groups: mean (standard deviation). The male groups: median (Q1, Q3). *^w^*
*Water from foods and beverages.* One-way ANOVA with Tuckey post-hoc tests were used for comparing females across groups. ^a^ Kruskal–Wallis with Dunn post-hoc tests were used for comparing males across groups due to the small sample size of males. ^b^ The male group comparison did not show a significant difference. Statistically significant values are written bold.

**Table 3 nutrients-12-00055-t003:** Intake of selected vitamins, minerals, and trace elements by gender and to the time in WFPB lifestyle program: short (0.5–<2 years), medium (2–<5 years), and long-term (5–10 years) (groups (1, 2 and 3)).

	Group 1 (*n* = 51)		Group 2 (*n* = 56)		Group 3 (*n* = 44)		*p*-Value
Micronutrients (Per Day)	Female	Male	Female	Male	Female	Male	F/M
**Vitamins**							
Thiamine (mg)	2.7 (1.1)	3.1 (2.7, 3.7)	2.8 (0.8)	3.5 (2.9, 3.7)	3.2 (2.1)	3.5 (3.3, 3.9)	0.199/0.460
Riboflavin (mg)	2.5 (1.6)	2.1 (1.8, 3.3)	3.1 (1.6)	2.4 (2.1, 3.7)	3.5 (2.2)	2.2 (2.0, 4.6)	0.083/0.676
Niacin (mg)	28 (8)	34 (26, 36)	30 (7)	33 (28, 39)	31 (10)	34 (31, 45)	0.286/0.787
Pantothenic acid (mg)	10 (3)	12 (10, 12)	11 (3)	12 (10, 15)	11 (4)	13 (11, 15)	0.381/0.676
Vitamin B_6_ (pyridoxine) (mg)	3.7 (1.2)	4.7 (3.4, 5.0)	3.7 (1.0)	4.5 (3.4, 5.4)	4.2 (1.8)	4.9 (3.9, 5.5)	0.141/0.857
Biotin (µg)	93 (33)	106 (84, 127)	98 (37)	96 (74, 126)	111 (48)	116 (86, 134)	0.169/0.897
Folate (µg folate equivalent)	757 (215)	925 (705, 1033)	761 (210)	887 (810, 1199)	772 (219)	798 (729, 1016)	0.959/0.570
Vitamin B_12_ (µg)	218 (270)	287 (4, 361)	230 (319)	288 (144, 430)	304 (375)	288 (287, 289)	0.501/0.370
Vitamin C (mg)	346 (230)	331 (247, 431)	322 (97)	279 (261, 410)	336 (154)	319 (248, 413)	0.822/0.891
Retinol equivalents *^re^* (mg)	3.7 (1.3)	3.9 (3.1, 4.8)	3.4 (1.6)	3.6 (3.1, 4.4)	3.8 (2.9)	4.6 (2.8, 5.4)	0.726/0.690
Vitamin D (µg)	8.4 (5.9)	5.9 (3.8, 10.2)	13.1 (14.4)	5.4 (4.1, 8.0)	12.9 (8.8)	7.7 (4.8, 12.3)	0.118/0.496
Vitamin E (mg)	26 (12)	29 (20, 33)	28 (10)	22 (21, 25)	30 (14)	29 (24, 36)	0.462/0.582
**Minerals**							
Calcium (mg)	1012 (313)	1155 (887, 1401)	991 (213)	1173 (1021, 1475)	1054 (361)	1235 (981, 1531)	0.659/0.754
Magnesium (mg)	902 (311)	987 (835, 1220)	801 (223)	883 (804, 1287)	819 (351)	1086 (833, 1145)	0.294/0.914
Phosphorus (mg)	1714 (434)	2189 (1720, 2489)	1623 (402)	2201 (1889, 2542)	1662 (404)	2382 (1792, 2556)	0.632/0.762
Potassium (mg)	4759 (1232)	6177 (4074, 6605)	4329 (1220)	6235 (5229, 6835)	4635 (1394)	5968 (5120, 6845)	0.309/0.761
Sodium (mg)	1956 (921)	2160 (1563, 3260)	2013 (1054)	1729 (1492, 2366)	1875 (779)	1945 (1806, 2684)	0.824/0.558
Chloride (mg)	3181 (1423)	3580 (2703, 4890)	3208 (1664)	2781 (2597, 4024)	3070 (1193)	3179 (2876, 4748)	0.918/0.500
**Trace Elements**							
Iron (mg)	38 (31)	40 (32, 46)	33 (7)	38 (36, 46)	34 (10)	40 (36, 42)	0.528/0.984
Copper (mg)	3.8 (1.0)	5.2 (4.1, 5.9)	3.7 (1.1)	4.3 (3.1, 5.4)	3.8 (1.1)	4.9 (4.1, 5.8)	0.831/0.679
Iodine (µg)	215 (75)	238 (165, 290)	234 (69)	238 (198, 282)	243 (82)	253 (196, 312)	0.295/0.717
Zinc (mg)	19 (6)	24 (18, 26)	19 (4)	21 (19, 24)	19 (6)	24 (21, 27)	0.897/0.819
Chrome (µg)	82 (27)	77 (53, 104)	77 (28)	77 (50, 110)	82 (38)	74 (49, 81)	0.743/0.711
Molybdenum (µg)	106 (80)	68 (48, 115)	86 (51)	86 (39, 125)	90 (56)	58 (42, 100)	0.363/0.780
Selenium (µg)	110 (49)	127 (119, 150)	114 (43)	117 (85, 147)	111 (41)	120 (107, 136)	0.931/0.538

The female groups: mean (standard deviation). The male groups: median (Q1, Q3). *^re^* Retinol equivalents = vitamin A + α-carotene (1 mg retinol equivalent = 12 mg α-carotene) + β-carotene (1 mg retinol equivalent = 6 mg β-carotene) + γ-carotene (1 mg retinol equivalent = 12 mg γ-carotene). One-way ANOVA was used for comparing females across groups. The Kruskal–Wallis test was used for comparing males across groups.

**Table 4 nutrients-12-00055-t004:** Intake of food groups among participants in WFPB lifestyle program, and comparison with two studies on PB dieters.

Country	Slovenia (Present Study)				Finland [16]	France [66]
Year of Publication	2019				2016	2017
Food Groups (g Per Day)	Whole Sample (*n* = 151)	Females (*n* = 109)	Males (*n* = 42)	*p*-Values	Whole Sample (*n* = 22)	Whole Sample (*n* = 789)
**Foods of Vegetable Origin**				F vs. M		
Vegetables (unprocessed)	455 (190)	433 (175)	510 (219)	0.040	277	366.2
Fruits (unprocessed)	363 (187)	358 (179)	377 (209)	0.618	254	364
Grain and products	178 (114)	147 (90)	257 (131)	<0.001	248 ^a^	232 ^c^
Legumes	166 (115)	143 (94)	226 (140)	<0.001	156 ^b^	73.2
Potatoes	140 (123)	130 (115)	165 (139)	0.217	151	58
Nuts and seeds	52 (46)	44 (29)	74 (69)	0.042	11	52.6 ^d^
Bread and bakery products	43 (50)	40 (43)	50 (64)	0.896	-	-
PB Meal replacement	43 (72)	48 (79)	28 (50)	0.145	-	-
Spices and herbs	32 (40)	35 (44)	24 (24)	0.054	-	-
Fruits (processed)	20 (33)	18 (27)	25 (44)	0.687	137 ^f^	-
Pasta	17 (35)	12 (25)	29 (50)	0.163	25	-
Fast food and ready meals	6 (34)	4 (18)	12 (58)	0.505	-	-
Vegetables (processed)	5 (23)	5 (24)	4 (20)	0.860	-	-
Sweet products	3 (11)	3 (13)	1 (4)	0.577	20	108.3 ^e^
Alcohol drinks	1 (12)	2 (14)	0.0	0.279	-	88.8
Vegetable oil and fat	1 (4)	1 (4)	2 (5)	0.409	51 ^h^	14.5
Sweeteners	0.2 (0.8)	0.1 (0.7)	0.2 (1.1)	0.524	-	-
**Foods of Animal Origin**						
Sea fish and products	3 (18)	2 (13)	6 (26)	0.359	0	12.8
Red meat	0.6 (0.6)	0.3 (2.3)	1.4 (9.2)	0.817	0	18.3
White meat	0.5 (5.3)	0.7 (5.5)	0.0	0.378	0	-
Meat product	0.2 (2.3)	0.3 (2.7)	0.0	0.378	0	5.8
Dairy products	0.1 (3)	0.5 (3.4)	0.3 (2.1)	0.703	7 ^g^	45
Milk	0.1 (1.6)	0.2 (1.9)	0.0	0.535	59 ^g^	-
Eggs and products	0.0	0.0	0.0	NaN	-	5.4
Animal fats	0.0	0.0	0.0	NaN	-	9.5

Data are means (standard deviation). NaN = not a number. A *T*-test was used to compare differences between genders. ^a^ Combined rye flour products (84 g/day), whole grain (139 g/day) and rice (25 g/day). ^b^ Combined legumes (81 g/day), tofu and soy flour (68 g/day) and soybeans (7 g/day). ^c^ Combined quinoa, corn, and other cereals group (27.3 g/day), refined cereals and starchy foods group (122.9 g/day) and whole starch food group (83.4 g/day). ^d^ Combined uncooked cereals and seed group (13 g/day), nuts group (19.6 g/day) and germinated seeds group (20 g/day). ^e^ Combined cookies and diet biscuit enriched with cereals group (6.4 g/day), salty snacks and biscuits group (8.1 g/day) and sweet and fatty foods (93.8 g/day). ^f^ Processed fruits include fruit and berry juices group (103 and 34 g/day). ^g^ Included soy beverages, soy yoghurt, and groats. ^h^ Vegetable fat included margarine, oils and other fats (dressing and mayonnaise).

**Table 5 nutrients-12-00055-t005:** Cardiovascular health (lipids and BP) and safety marker status (serum UA and haemoglobin) according to the time in WFPB lifestyle program: short (0.5–<2 years), medium (2–<5 years) and long-term (5–10 years) (groups 1, 2 and 3), and by gender.

Parameter	Group 1 (*n* = 51)		Group 2 (*n* = 56)		Group 3 (*n* = 44)		*p*-Value
Gender (F/M)	Female	Male	Female	Male	Female	Male	F/M
**Laboratory Variables**							
S-cholesterol (mmol/L)	3.7 (0.8)		3.8 (0.8)		3.5 (0.8)		0.231 *
	3.9 (0.8)	3.3 (2.7, 3.9)	3.9 (0.8)	3.7 (3.1, 4.2)	3.7 (0.7)	2.7 (2.6, 3.4)	0.495/0.283
LDL-cholesterol (mmol/L)	2.1 (0.6)		2.2 (0.7)		1.8 (0.6)		0.010
	2.2 (0.6)	1.9 (1.5, 2.4)	2.1 (0.7)	2.2 (1.9, 2.6)	1.8 (0.5)	1.6 (1.4, 1.9)	0.025/0.233
HDL-cholesterol (mmol/L)	1.4 (0.4)		1.4 (0.3)		1.4 (0.4)		0.391 *
	1.5 (0.4)	1.1 (1.0, 1.2)	1.5 (0.3)	1.1 (1.0, 1.2)	1.6 (0.3)	1.1 (1.0, 1.2)	0.233/0.988
Triglycerides (mmol/L)	0.9 (0.4)		0.8 (0.3)		0.8 (0.4)		0.648 *
	0.8 (0.3)	0.9 (0.7, 1.4)	0.8 (0.3)	0.9 (0.7, 1.1)	0.8 (0.3)	0.8 (0.6, 0.9)	0.933/0.681
S-Uric Acid (μmol/L)	288 (70)		266 (67)		264 (65)		0.124 *
	257 (51)	352 (312, 385)	238 (41)	371 (331, 387)	240 (59)	304 (294, 362)	0.194/0.065
Haemoglobin (g/L)	141 (12)		142 (13)		142 (11)		0.850 *
	135 (9)	154 (150, 157)	137 (11)	156 (150, 162)	139 (10)	146 (143, 157)	0.191/0.160
**Blood Pressure (mmHg)**							
Systolic	116 (11)		114 (11)		115 (10)		0.832 *
	114 (12)	119 (111, 123)	112 (11)	121 (117, 124)	113 (10)	117 (115, 127)	0.697/0.674
Diastolic	70 (8)		72 (10)		72 (7)		0.384 *
	70 (8)	71 (64, 76)	71 (11)	76 (72, 79)	72 (6)	75 (69, 76)	0.615/0.220

The female groups: mean (standard deviation). The male groups: median (Q1, Q3). * Listed *p*-values are for interactions between group 1, 2 and 3. One-way ANOVA was used for comparing females across groups. The Kruskal Wallis test was used for comparing males across groups. The Tukey post-hoc test was performed when differences were statistically significant. For five participant that were >65 years of age for the table transparency purpose we have used the same reference values as for adults up to 65 years of age since blood pressure of all these older participants were below 140/90 mmHg (from 127/75 to 134/87 mmHg). Statistically significant values are written bold.

**Table 6 nutrients-12-00055-t006:** Percentage of participants in the WFPB lifestyle program (0.5–10 years), by gender, achieving recommendations [61,62,67,68] for cardiovascular risk factors and safety markers.

Parameter	Whole Sample (*n* = 151)		
Gender (F/M)	Female (*n* = 109)	Male (*n* = 42)	*p*-Value
Hypothesis (Normal/High)			F vs. M
**LDL-Cholesterol**	<3 mmol/L [62]		
Normal	140 (93%)		
High	11 (7%)		
Normal	102 (94%)	38 (91%)	0.500
High	7 (6%)	4 (9%)	0.500
**Triglycerides**	<1.7 mmol/L [62]		
Normal	146 (97%)		
High	5 (3%)		
Normal	108 (99%)	38 (91%)	0.021
High	1 (1%)	4 (9%)	0.021
**Blood Pressure**	≤129/84 mmHg [61]		
Systolic	≤129 mmHg		
Normal	133 (88%)		
High	18 (12%)		
Normal	100 (92%)	33 (79%)	0.046
High	9 (8%)	9 (21%)	0.046
Diastolic	≤84 mmHg		
Normal	143 (95%)		
High	8 (5%)		
Normal	101 (93%)	42 (100%)	0.107
High	8 (7%)	0 (0.0%)	0.107
**S-Uric Acid**	Female: 154–357 mmol/L; Male: 308–428 mmol/L [67]		
Normal	132 (87%)		
High	5 (3%)		
Low	14 (9.3%)		
Normal	106 (97%)	26 (62%)	<0.001
High	1 (1%)	4 (9%)	<0.001
Low	2 (2%)	12 (29%)	<0.001
**Haemoglobin**	Non pregnant female: >120 g/L; Male: >130 g/L [68]		
Normal	146 (97%)		
High	5 (3%)		
Normal	104 (95%)	42 (100.0%)	0.323
Low	5 (5%)	0 (0.0%)	0.323

Data are means. For LDL-cholesterol, triglycerides, blood pressure and haemoglobin: Normal = the number and percentage of participants (whole sample and female or male) who are within reference, High = the number and percentage of participants that are above the reference. For S-Uric Acid: Normal = the number and percentage of participants (whole sample, female or male) who are within reference, High = the number and percentage of participants who are above the reference, Low = the number and percentage of participants who are below the reference. A T-test was used to compare differences between genders.

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
