# Peer review of "Dietary Intakes and Cardiovascular Health of Healthy Adults in Short-, Medium-, and Long-Term Whole-Food Plant-Based Lifestyle Program"

_nutrients, 2019, doi:10.3390/nu12010055_

Round 1

Reviewer 1 Report

This is a well designed study. I have a number of issues with the study.

Diet and supplements- since the subjects received supplementation, we can't make conclusions regarding the merit of PBD on its own. I couldn't find the source for the EPA/DHA.IF it;s fish, it should be indicated. Source should be detailed, regardless, unless it is and I somehow missed it. Results- this is the least significant part- we know that WPBD will lower cholesterol and BP. What would be more interesting is to see the changes omr baseline diet and the length of time it took. Lipids- differentiation like Small particle LDL, LP9a),  as well as CRP can be of value. IF blood samples were frozen, it can be done, in which case I would recommend also following on Gaectin-3 .

Author Response

Comments and suggestions for authors

REVIEWER 1

This is a well designed study. I have a number of issues with the study.

Thank you. 

1.)     Diet and supplements- since the subjects received supplementation, we can't make conclusions regarding the merit of PBD on its own.

We fully agree. In the manuscript, we have emphasized that the benefits are not solely due to PBD, but due to their lifestyle.

We have already written about lifestyle in:

-          Title: “Whole-Food Plant-Based Lifestyle Program”;

-          2. Materials and Methods, 2.3. Intervention: WFPB lifestyle program: “The WFPB lifestyle program included the nutritional part, the component of physical activity (PA) and the support system (see Table 1).”

-           4. Discussion, 4.4. Support system: “The essence of the support system is to optimally support individuals or groups to follow the proposed lifestyle change and maintain a long-term, healthy and active lifestyle as “the new normal”. The support system is essential for adherence to lifestyle change.”

-          5. Strengths and limitation: “Additionally, our results were not limited to diet only, since the participants started a healthy and active lifestyle when entering the supplemented WFPB diet lifestyle and regular PA is known to be associated with lower BMI and BF percentage [84], especially in combination with a weight-management dietary program [85]. Low and moderate intensity PA is also associated with the reduction of total, but not LDL-cholesterol [86]. An additional stimulus for greater success of the supplemented WFPB diet was the extensive support system [41].”

-          However, MR that was used ≤ 10% of energy and other supplements (except EPA and DHA, fish oils) were plant-based derived. We are well aware that the use of supplements represents a deviation from the strict WFPB diet (but not PBD). They were used in order to facilitate the implementation of the program and to ensure an adequacy of all micronutrients. In a separate paper (submitted to Nutrients), we evaluated separately and combined the nutrient intake from foods as well as from meal replacement (MR) and supplements and compared it with the dietary recommendations for both genders.

2.)         I couldn't find the source for the EPA/DHA.IF it’s fish, it should be indicated. Source should be detailed, regardless, unless it is and I somehow missed it.

The source of EPA and DHA is fish oil. It was written below Table 1.

3.)         Results- this is the least significant part- we know that WPBD will lower cholesterol and BP. What would be more interesting is to see the changes or baseline diet and the length of time it took.

We added the following text to the sub-chapter 2.2. Subject: “The Western-type diet of the participants at baseline included foods of animal origin (milk and dairy products, meat and meat products, eggs and egg products, fish), refined grains and flour (i.e., white bread, pasta), sweets and pastry, animal fat and vegetable oils, sugar-sweeten beverages as well as alcoholic beverages. It contained much less fresh vegetables and fruits, while unrefined and whole plant grains were largely absent. In short, this resulted in a high intake of fat, especially SFA and cholesterol, and a low intake of PUFA. It also contained free sugars and more alcohol, while the fiber intake was much lower.”

The average duration on WFPB lifestyle for each group (1, 2 and 3) was written in sub-chapter 3.1. Characteristics of the participants: “The average duration on WFPB lifestyle program for all participants was 4.1 years (1.3 years for group 1, 3.9 years for group 2 and 7 years for group 3)”.

4.)         Lipids- differentiation like Small particle LDL, LP9a), as well as CRP can be of value. IF blood samples were frozen, it can be done, in which case I would recommend also following on Gaectin-3.

We are well aware of this. Unfortunately, we had very limited research resources. We also do not have frozen blood samples of the participants.

Reviewer 2 Report

I admit that the authors carried out lots of effort to investigate the influence of the short-, medium- and long-term WFPB lifestyle program on dietary intake and CV heath status. But overall, there are too many contents that are irrelevant to the purpose of this manuscript, and thus this makes it difficult to understand what the purpose of this study is. Also, all the tables are not well-formed, making it difficult to understand the contents of the tables.

Please revise the following points.

1. Page 2: Add reasons for making a hypothesis(H2) in the introduction; (H2: There is difference in lipid profile and BP status between the short- and medium-term, but not medium- and long-term groups (by gender))

2. page 3: The explanation for Intervention (the WFPB lifestyle program) is too long. The purpose of this study is not to determine the effect of intervention. This study is a cross-sectional study. However, the section describing the intervention is so long that it interferes with the flow of this paper. Since the authors presented Table 1 as the summery of WFPB lifestyle program, It is recommended that you reduce the intervention description to less than one page. In the original manuscript, the explanation of the intervention takes three pages.

3. Page 7: The characteristics of the study subjects is written in this manuscript without a table attached.

Please add an additional table (Characteristics of the participants) in this paper. Otherwise, it would be better to add the exploration of the subjects related to this study such as age and gender, and the average duration on WFPB lifestyle program in Table 1S (supplementary file). Fig. 1 has already been cited in the selection of subjects in the research method, and it is appropriate to be in the method, not the results part. Thus move the content about Fig.1 to the method section (the selection of subjects)s Of the explanation of the characteristics of the subjects, BMI, muscle mass, physical activity, and PSQI differences of the subjects in page 7 are not relevant with this study. So either present table or erase this part.

4. Need to modify all the tables presented in this paper to the proper format of this journal.

Lines in the table 2, ± in table 3 Need footnote for NS, NaN in table 6 & % E, in table 2, All tables need proper footnotes for the description of the statistics (what statistics applied for p-value. No display of results for the post-hoc test. For example, in table 2, free sugars (%E), PUFA, linoleic acid, α-linolenic acid for females and water for males, table 5, LDL,) Page 8: The p-values presented in the table differ from those described in the results (Check free sugars (%E), PUFA, linoleic acid, α-linolenic acid for females and water for males see in Tables 2 and 3) In table 6 (page 15); need footnotes of each reference values for such an division (like normal whole or high whole but 3 divisions of uric acid) for all parameters in table 6

5.  P21 (4.4)- Delete the description of the support system from the discussion It is irrelevant to the purpose of this paper and rather hinders the flow of the paper.

Author Response

REVIEWER 2

I admit that the authors carried out lots of effort to investigate the influence of the short medium- and long-term WFPB lifestyle program on dietary intake and CV heath status. But overall, there are too many contents that are irrelevant to the purpose of this manuscript, and thus this makes it difficult to understand what the purpose of this study is. 

Yes, we agree with this and have corrected it. We deleted a lot of text (please see the revised manuscript with the tracked changes on the side).

Also, all the tables are not well-formed, making it difficult to understand the contents of the tables.

We corrected them, i.e., improved the shape, lines and information, added statistics and explanation in the footnotes, deleted/added redundant abbreviations, p – value alignment).

Page 2: Add reasons for making a hypothesis(H2) in the introduction; (H2: There is difference in lipid profile and BP status between the short- and medium-term, but not medium- and long-term groups (by gender)).

We added the following explanation just after H2 in “1. Introduction”: »According to the our clinical experiences and several randomized control trials that have utilized WFPB diet (Barnard et al., 2009; Wright, Wilson, Smith, Duncan, & McHugh, 2017), we assumed that all the benefits for a lipid profile and BP status would be achieved already within the first two years.”

Page 3: The explanation for Intervention (the WFPB lifestyle program) is too long. The purpose of this study is not to determine the effect of intervention. This study is a cross-sectional study. However, the section describing the intervention is so long that it interferes with the flow of this paper. Since the authors presented Table 1 as the summery of WFPB lifestyle program, It is recommended that you reduce the intervention description to less than one page. In the original manuscript, the explanation of the intervention takes three pages.

We shortened the description of the intervention to less than one page (please see the revised manuscript with the tracked changes on the side).

Page 7: The characteristics of the study subjects is written in this manuscript without a table attached. Please add an additional table (Characteristics of the participants) in this paper. Otherwise, it would be better to add the exploration of the subjects related to this study such as age and gender, and the average duration on WFPB lifestyle program in Table 1S (supplementary file). Fig. 1 has already been cited in the selection of subjects in the research method, and it is appropriate to be in the method, not the results part. Thus move the content about Fig.1 to the method section (the selection of subjects).

The characteristics of the study subject were already in the supplementary Table 1S. The average duration of a WFPB lifestyle was written in sub-chapter “3.1. Characteristics of the participants” of the “3. Results” section.

We have moved the Figure 1 in the method section (2.2. Subject), at the end of paragraph: “Thus, we included 151 adult participants (91% of initially included, Figure 1), aged 18-80 years in the final analysis.”

Of the explanation of the characteristics of the subjects, BMI, muscle mass, physical activity, and PSQI differences of the subjects in page 7 are not relevant with this study. So either present table or erase this part.

We agree that the explanation of the lifestyle characteristics of the subject is too long for the purpose of this study. That being so, we deleted some parts and shortened others to a minimum as suggested (we have too many tables to add additional ones), but we consider that basic information that we think are valuable for the results obtained in Discussion chapter. We shortened the characteristics of the subject (BMI, PA and lifestyle factors) to a minimum (chapter “3.1.”): “All participants improved their baseline mean pre-obesity BMI range to a normal BMI range (from 26.4 to 23.9 kg/m2), and experienced decreased BM and BF % points (-7.1 kg and -6.4% points, respectively) (p < 0.001 for all). Participants were physically very active (International Physical Activity Questionnaire (L-IPAQ) average score: 5542 METs min/week), had good sleep quality (Pittsburgh Sleep Quality Index (PSQI) average score: 2.7) and perceived low stress (Perceived Stress Questionnaire (PSQ) average score: 0.3).

Need to modify all the tables presented in this paper to the proper format of this journal.

We have corrected the format of the Tables.

Corrected: Lines in the table 2  % E, in table 2,  ± in table 3

Added: Footnotes for NS, NaN in the table 6;

Added (please see below): All tables need proper footnotes for the description of the statistics (what statistics applied for p-value.

No display of results for the post-hoc test. For example, in table 2, free sugars (%E), PUFA, linoleic acid, α-linolenic acid for females and water for males, table 5, LDL,).

Macronutrients: We added the following paragraph) in chapter “3.2.”: Dietary and food intakes of participants in our WFPB lifestyle program“ (and was already added in Discussion section): “The comparison between genders according to different groups showed that, with the female participants, group 1 had a statistically significant lower relative intake of free sugars (% E) than group 3 (p = 0.032), while it showed a higher absolute intake of PUFA (p = 0.017) and linoleic acid (p = 0.041) compared to group 2 as well as a higher absolute intake of alpha-linoleic acid compared to group 2 (p = 0.050) and group 3 (p = 0.046). With the male participants, group 3 had a statistically significant higher intake of water compared to group 2 (p = 0.046) and group 1 (p = 0.041). The comparison between genders of the three groups did not show any significant difference when it came to the intake of micronutrients.”

LDL-cholesterol: We added a p-value in the results section (“3.3.2. Main findings”): “Contrary to the set hypothesis, females in group 3 had a statistically significant lower LDL-cholesterol compared to group 1 (p = 0.034).”

Page 8: The p-values presented in the table differ from those described in the results (Check free sugars (%E), PUFA, linoleic acid, α-linolenic acid for females and water for males see in Tables 2 and 3)

We (and original statistics with statisticians) double checked the p-values in Tables 2 and 3 and other tables and in the results. No mistakes were found. In Table 2S, we found a significant difference in almost all nutrients (female vs. male). In this table, the results and the p-values are also correct.

In table 6 (page 15); need footnotes of each reference values for such an division (like normal whole or high whole but 3 divisions of uric acid) for all parameters in table 6.

We deleted “whole” since it is already written at the top of Table 6 (“Whole sample”). We added the explanations for Normal, High, Low (S-UA).

Reviewer 3 Report

In this study by authors B Jakse et al, 151 adults were recruited to participate in a whole food plant based (WFPB) lifestyle program for short (0.5-2 years), medium (2-5 years) and long-term (5-10 years) and cardiovascular risk factors, such as LDL-cholesterol, HDL-cholesterol, total cholesterol, triglyceride, and systolic blood pressure were measured. Authors concluded that WFPB lifestyle is favorable for cardiovascular health based on the guideline recommended targets (The European Society of Cardiology (ESC) and the European Atherosclerosis Society (EAS)) for LDL-cholesterol and triglyceride. This is a long term investigation of the association between plant based diet and cardiovascular health, which provides some convincing evidence. However, I have some concerns, which were listed below.

Since there are a lot of traditional cardiovascular risk factors, authors should provide more laboratory parameters, such as glomerular filtration rate, fasting glucose level;

In Table 2, it seems that a lot of P values for the parameter comparison between males and females should be less than 0.05. In addition, why the authors labeled the p values which are smaller than 0.05 as “NS”. Authors had better double-check all the P values listed in the Tables;

Authors had better listed all abbreviations used in this manuscript before introduction part.

Author Response

REVIEWER 3

Since there are a lot of traditional cardiovascular risk factors, authors should provide more laboratory parameters, such as glomerular filtration rate, fasting glucose level

We are well aware of this. Unfortunately, we were very limited by research resources.

In Table 2, it seems that a lot of P values for the parameter comparison between males and females should be less than 0.05. In addition, why the authors labeled the p values which are smaller than 0.05 as “NS”. Authors had better double-check all the P values listed in the Tables.

We double checked all the p-values and they were correct. In Table 2, we compared the energy and macronutrients intakes by gender and by the duration on the WFPB lifestyle (F/M), not between the genders which is in the Table 2S, were have found the statistically significant differences for almost all macronutrients. True. We labeled only p-values that were smaller than 0.05 as NS (non-significant), as was suggested.

We added the following text below the tables:

Table 2:

“NS = non-significant. One-way ANOVA with Tuckey post-hoc tests were used for comparing females across groups. Kruskal Wallis with Dunn post-hoc tests were used for comparing males across groups due to a small sample size of males. *The male group comparison did not show a significant difference.”

Table 2S:

“NS = non-significant. T-test was used to compare differences between genders.”

Table 3:

“reRetinol equivalents = vitamin A + α-carotene (1 mg retinol equivalent = 12 mg α-carotene) + β-carotene (1 mg retinol equivalent = 6 mg β-carotene) + γ-carotene (1 mg retinol equivalent = 12 mg γ-carotene). NS = non-significant. One-way ANOVA was used for comparing females across groups. The Kruskal Wallis test was used for comparing males across groups.”

Table 3S:

“reRetinol equivalents = vitamin A + α-carotene (1 mg retinol equivalent = 12 mg α-carotene) + β-carotene (1 mg retinol equivalent = 6 mg β-carotene) + γ-carotene (1 mg retinol equivalent = 12 mg γ-carotene). NS = non-significant. A T-test was used to compare differences between genders.”

Table 4:

“NS = non-significant; NaN = not a number. A T-test was used to compare differences between genders.”

Table 5:

“NS = non-significant. One-way ANOVA was used for comparing females across groups. The Kruskal Wallis test was used for comparing males across groups. The Tukey post-hoc test was performed when differences were statistically significant.”

Table 5S:

“NS = non-significant. A T-test was used to compare differences between genders.”

Table 6:

“NS = non-significant. For LDL-cholesterol, triglycerides, blood pressure and haemoglobin: Normal = the number and percentage of participants (whole sample and female or male) who are within reference; High = the number and percentage of participants that are above the reference. For S-Uric Acid:  Normal = the number and percentage of participants (whole sample, female or male) who are within reference; High = the number and percentage of participants who are above the reference; Low = the number and percentage of participants who are below the reference. A T-test was used to compare differences between genders.

We added the abbreviations below the Conclusion:  

“3-DR, three-day weighted dietary record

BM, body mass

BMI, body mass index

BP, blood pressure

CVD, cardiovascular disease

DHA, docosahexaenoic acid (C22:6n-3)

EPA, (eicosapentaenoic acid (C20:5n-3)

EAS, European Atherosclerosis Society

ESC, European Society of Cardiology

FFQ, Food Frequency Questionnaire

FU, follow up

H1, hypothesis one

H2, hypothesis two

H3, hypothesis three

MR, plant-based meal replacement

OPEN, Open Platform for Clinical Nutrition

PA, physical activity

PBD, plant-based diet

UA, uric acids

WFPB, whole-food plant-based”.
